# Brain Magnetic Resonance Findings in 117 Children with Autism Spectrum Disorder under 5 Years Old

**DOI:** 10.3390/brainsci10100741

**Published:** 2020-10-16

**Authors:** Magali Jane Rochat, Giacomo Distefano, Monica Maffei, Francesco Toni, Annio Posar, Maria Cristina Scaduto, Federica Resca, Cinzia Cameli, Elena Bacchelli, Elena Maestrini, Paola Visconti

**Affiliations:** 1IRCCS Istituto delle Scienze Neurologiche di Bologna, Diagnostica Funzionale Neuroradiologica, Ospedale Bellaria di Bologna, 40139 Bologna, Italy; 2Dipartimento di Medicina Specialistica Diagnostica e Sperimentale (DIMES), Università di Bologna, 40126 Bologna, Italy; giacomo.distefano@studio.unibo.it; 3IRCCS Istituto delle Scienze Neurologiche di Bologna, UOC Neuroradiologia, Ospedale Bellaria di Bologna, 40139 Bologna, Italy; monica.maffei@isnb.it (M.M.); francesco.toni@isnb.it (F.T.); 4IRCCS Istituto delle Scienze Neurologiche di Bologna, UOSI Disturbi dello Spettro Autistico, Ospedale Bellaria di Bologna, 40139 Bologna, Italy; annio.posar@unibo.it (A.P.); m.scaduto@isnb.it (M.C.S.); f.resca@ausl.bologna.it (F.R.); 5Dipartimento di Scienze Biomediche e Neuromotorie (DIBINEM), Università di Bologna, 40126 Bologna, Italy; 6Dipartimento di Farmacia e Biotecnologie, Università di Bologna, 40126 Bologna, Italy; cinzia.cameli3@unibo.it (C.C.); elena.bacchelli@unibo.it (E.B.); elena.maestrini@unibo.it (E.M.)

**Keywords:** magnetic resonance imaging, autism spectrum disorder, incidental findings, neurobiology

## Abstract

We examined the potential benefits of neuroimaging measurements across the first 5 years of life in detecting early comorbid or etiological signs of autism spectrum disorder (ASD). In particular, we analyzed the prevalence of neuroradiologic findings in routine magnetic resonance imaging (MRI) scans of a group of 117 ASD children younger than 5 years old. These data were compared to those reported in typically developing (TD) children. MRI findings in children with ASD were analyzed in relation to their cognitive level, severity of autistic symptoms, and the presence of electroencephalogram (EEG) abnormalities. The MRI was rated abnormal in 55% of children with ASD with a significant prevalence in the high-functioning subgroup compared to TD children. We report significant incidental findings of mega cisterna magna, ventricular anomalies and abnormal white matter signal intensity in ASD without significant associations between these MRI findings and EEG features. Based on these results we discuss the role that brain MRI may play in the diagnostic procedure of ASD.

## 1. Introduction

Autism spectrum disorder (ASD) is a neurodevelopmental disorder characterized by various degrees of impaired social communication and a narrow range of interests and activities. Symptoms occur in early childhood and tend to persist into adulthood with consistent impact on the individual’s daily functioning [1].

Epidemiologic surveys indicate a striking worldwide increase in ASD prevalence over the last few decades [2]: the latest update from the Centers for Disease Control and Prevention (CDC) of the United States estimates a prevalence of 1 out of 54 children [3]. The increase in autism prevalence represents one of the most acute public health crises in recent decades, with an economic impact stressing healthcare systems worldwide [4,5].

It is well established that genetic factors strongly contribute to the risk of ASD, which manifests an extreme degree of clinical and genetic heterogeneity. Despite the high heritability, with estimates as high as 90%, only about 15%–30% of ASD individuals have an identifiable genetic cause such as chromosomal abnormalities, submicroscopic copy number variations, and highly penetrant single gene mutations [6]. The genetic etiology underlying the majority of cases remains still unknown.

Advances in electroencephalogram (EEG) and neuroimaging research suggest the existence of functional and neuroanatomical differences in the brain of individuals with ASD that are discernible very early in the first year of life [7,8]. Epileptiform abnormalities are commonly reported among ASD children, with frequencies ranging from 10% to 72% [9,10], while structural magnetic resonance imaging (MRI) conducted in young children with ASD assessed the presence of brain volume overgrowth [11,12], aberrant pattern of cortical thickness [13] and increased volume of cerebrospinal fluid in the subarachnoid space [14,15]. Moreover, connectivity studies constantly report a disruption of brain connectivity in toddlers with ASD [16].

Brain MRI frequently detect the presence of incidental findings (IFs) in ASD children. IFs are defined as “unsought asymptomatic brain findings generated while seeking other information of interest” [17]. The rate of IFs in healthy children is around 16.4% [18]. Intracranial cysts are the most frequently found IFs (10.2%), followed by nonspecific white matter hyperintensities (1.9%), Chiari I malformation (0.8%) and intracranial neoplasms (0.2%). Several neuroimaging studies have documented an elevated prevalence rate of IFs in children and adolescents with ASD [17,19,20,21,22,23], which could suggest a condition of neural maldevelopment underlying the ASD. However, in contrast to the cited studies, other findings suggested that ASD diagnosis was not significantly related to higher rates of IFs nor to specific brain findings [24,25]. Although IFs’ pathogenic contribution is rather questionable, their increased frequency in very young children with ASD is still a matter of debate.

Our study aims to evaluate the effective potential of conventional MRI in unveiling early morphologic biomarkers for ASD, including IFs, in a large group of children with ASD, controlling for sample homogeneity in terms of age, developmental level, and diagnosis and compared with an age-matched control sample. The subgroup of children with MRI findings was analyzed for variation in a cognitive level, severity of autistic symptoms and the presence of EEG and genetic abnormalities.

## 2. Materials and Methods

From an initial cohort of 279 children with developmental disorders referred to the Center for ASD in the Child Neurology and Psychiatry Unit, IRCCS Institute of Neurological Sciences of Bologna from January 2011 to January 2020, 188 children received a diagnosis of ASD and underwent a brain MRI in order to complete a neurological workup (mean age 5.4 years, range 13 months–28.4 years; 153 males) specifically accomplishing a research project on the Genetics of ASD aimed at discovering correlations between genotype and clinical-neurological phenotype including sleep-awake EEG and MRI findings as well as standard metabolic testing, array-comparative genomic hybridization (array-CGH), and DNA analysis of MECP2 or Fragile-X syndrome (see the end of this paragraph). Exclusion criteria were as follows: defined genetic syndromes commonly associated with ASD (single gene disorders and microscopically visible chromosomal abnormalities [26]), prenatal infections including rubella and cytomegalovirus, neurometabolic or neurodegenerative diseases, severe prematurity or severe nutritional deficits and/or presence of sensory deficits, lack of parental consent to perform the MRI.

To obtain a more homogeneous sample, we restricted our final sample to children younger than 5 years of age at the time of the MRI (*n* = 117, mean age 3.4 years, range 13 months–4.11 years; 90 males).

All subjects were examined by a team of child neuropsychiatrists and neuropsychologists with certified experience in the diagnosis and neurobiological assessment of ASD. Children with ASD were diagnosed according to the Diagnostic and Statistical Manual of Mental Disorders (5th edition) [1] and confirmed by both Autism Diagnostic Observation Schedule-Second Edition [27] and Childhood Autism Rating Scale-Second Edition [28]. For children younger than 31 months at the time of scanning (19/117; 16.24%) a final diagnosis was confirmed when they reached at least 31 months (see Appendix A).

The cognitive functioning or developmental level was assessed using a battery of different neuropsychological tests chosen according to the chronological or mental age and verbal abilities (see Appendix A). Scores were normalized to z-scores for statistical analysis. 

Previous studies [20,21,22,23,24] demonstrated the existence of an IF prevalence among ASD individuals with intellectual disability. Therefore, in order to verify the presence of similar data in our sample of young children with ASD, two subgroups were defined on the basis of their cognitive or developmental level. The cognitive cut-off for intellectual disability was set according to both classical and more recent references (see Appendix A). Children presenting a global development (GD)/non-verbal intelligence quotient (IQ) of 70 or below were included in the group of ASD children with developmental delay (ASD-DD), while children presenting a GD/non-verbal IQ of 71 or higher were classified as high-functioning ASD (ASD-HF). MRI was also performed in 39 age-matched typically developing (TD) children (mean age: 3.2 years, range: 13 months–5.7 years; 21 males, 18 females). The control MRIs were retrospectively selected from our database of children with mild cranial trauma according to three criteria: (1) age matching with ASD children group, (2) same MRI scanning protocol as in children with ASD and (3) sufficient clinical information to ensure the lack of prior neurological or developmental disorders.

This study is part of a research project on the Genetics of ASD approved by the local Ethics Committee (Comitato Etico di Area Vasta Emilia Centro—CE-AVEC; code CE 14060). The data collected were part of the routine assessment of individuals suspected of having autism and parents signed an informed consent form to allow the scientific use of this information. All research was performed in accordance with the relevant guidelines and regulations.

### 2.1. Sleep-EEG Acquisition

The EEG was recorded on a SynAmpsNeuroScan Inc. (Herndon, VA, USA) system using Scan version 4.2. Applying the International 10–20 system, awake and sleep EEG recordings were performed in 85% (100/117) of the children with ASD, lacking parents’ consent in a minority of subjects.

### 2.2. Magnetic Resonance Imaging Acquisition

MR scans were made with patients under sedation to avoid movements during the acquisition of the images.

Scans of all ASD and TD children were acquired on two different clinical MRI scanners: a 1.5 Tesla (T) Siemens unit (*n* = 132) or a 3T General Electric MRI unit (GE HDxt, General Electric Healthcare, Milwaukee, WI, USA) (*n* = 24). No significant differences were found between 1.5 and 3T scanning machines yielding brain morphology assessment (in particular as concerns white matter evaluation). Namely, regardless of the machine used, the technical parameters applied (thickness, TE, TR) were set up to obtain high quality morphological sequences (T1, T2 and FLAIR T2 weighted, mainly) that provided adequate anatomical details and high contrast between tissues.

The standard imaging protocol always included: sagittal 3D T1-weighted images, completed with multi-planar reconstructions (repetition time [TR] = 2400 msec [1.5T] and 2300 msec [3T], inversion time [TI] 1000 msec [1.5T] and 850 msec [3T], echo time [TE] = 35 msec [1.5T] and 2.98 msec [3T], 1 mm isotropic voxel); axial TSE T2-weighted sequence (TR = 4000 msec [1.5T] 4720 msec [3T], TE = 98 msec [1.5T] and 111 msec [3T], 4 mm thickness), axial FLAIR T2-weighted sequence (TR = 9000 msec [1.5 and 3T], TE = 86 msec [1.5T] and 91 msec [3T], TI = 2500 msec [1.5T and 3T], 4 mm thickness) and axial 3D susceptibility weighted imaging—SWI (TR = 49 msec [1.5T] and 28 msec [3T], TE = 40 msec [1.5T] and 20 msec [3T], 2 mm thickness).

The management and clinical data of the patients were recovered using RIS^®^ system (Radiology Information Assistant). The images were collected using PACS^®^ archive (Picture Archiving and Communication System) and displayed on a reporting workstation. The images were assessed independently by two board certified pediatric neuroradiologists blinded to the diagnosis of each subject. Prior to reading the scans, the neuroradiologists established a list of predefined items to evaluate that were selected incorporating the main neuroradiologic findings reported in previous studies [17,21,23,24].

Particular attention was given to the presence of anomalies in the corpus callosum, cerebellum and brainstem, white matter signal intensity abnormalities, abnormal pattern of myelination, ventricular system size, Arnold Chiari I malformation, cortical dysplasia and atrophy, hippocampal malformations and pituitary gland abnormalities. These abnormalities were considered “major abnormal findings”. Another category, named MRI “minor abnormalities” included Virchow–Robin spaces dilatation, enlarged cisterna magna, pineal gland cysts, arachnoid or choroidal cysts that were considered as IFs even if not included in predefined categories. Plagiocephaly, cerebral tonsils displacement <5 mm, trigonocephaly and cystic pineal glands <6 mm were considered as normal variants.

Virchow–Robin spaces were classified as dilated when they were >3 mm, using the classification system established by Heier and colleagues [29]. Chiari malformation was confirmed only when a displacement ≥5 mm of one or both cerebellar tonsils through the foramen magnum was visible, as reported by Hildago et al. [30,31,32]. 

White matter lesions (single punctate, multiple punctate and confluent plaque-like) were visible on T2 and FLAIR sequences as signal hyperintensities. These lesions were further classified according to their location.

Ventricular dilatation (cut-off >10 mm) and asymmetric enlargement were also evaluated [33].

Pineal cysts were considered typical when unilocular, smooth edged, with homogeneous interior content in every MRI sequence and with a diameter of at least 6 mm [34].

The threshold of cisterna magna enlargement was >10 mm on midsagittal images [35].

The two independent neuroradiologists defined the scans as abnormal if any major and/or minor abnormal finding was present. On the contrary, scans were defined as normal if no brain findings or only normal variants were present. The consensus percentage for the neuroradiologists’ findings was of 93.16 % (Cohen’s Kappa= 0.86, 95% confidence interval: 0.87–0.97), which is considered an almost perfect agreement according to Landis and Koch classification [36]. More specifically, a 100% agreement was present for most IFs except for the following categories: *retrotrigonal WM* intensities (14/18, 77.78% consensus), *gliosis*, (9/10, 90%), *abnormal cortical development and/or organization* (1/2, 50%), *mega cisterna magna* (11/12, 91.67%), *pineal cyst* (12/13, 92.31%). A consensus agreement was reached when inconsistencies were found between the reviewers.

### 2.3. Statistical Analyses

A one-way analysis of variance (ANOVA) was used to examine age differences between ASD children and controls, as well as differences in autism severity between ASD-DD and ASD-HF subgroups. 

A chi-square test was used to compare the frequency of normal and abnormal MRI reports among ASD-HF, ASD-DD and TD subgroups.

A two-tailed Fisher’s exact test was used to examine sex differences between ASD children and controls as well as differences in the distribution of ASD-DD and ASD-HF individuals in three levels of symptoms. Fisher’s exact test was also used to study the frequency of MRI abnormalities in the DD and HF subgroups compared to the TD children group and to assess the distribution of IFs and to assess the distribution of IFs in the three levels of ASD symptoms categories and in children presenting EEG anomalies. 

All statistical analyses were conducted using the Statistica 8.0 Statsoft package, and *p* values of 0.05 or lower were considered as significant.

## 3. Results

In total, 156 ASD and TD children were enrolled for this study. One hundred and seventeen children presented with ASD, and 76 of them also presented with a developmental delay (ASD-DD), while 41 were classified as high functioning (ASD-HF). Thirty-nine were TD children. The age range was from 13 months to 5 years 7 months, without significant differences among the three groups (F(2151) = 1.5087, *p* = 0.224). Male subjects prevailed in ASD (90 males: 27 females) compared to the control group (21 males: 18 females) (Fisher’s Exact test = 0.008), while no significant difference in sex ratio was present between DD and HF autism groups (*p* = 0.498).

Autistic symptoms severity (ADOS-2 comparative scores) was significantly higher in the ASD-DD (M = 7.64, SE = 0.26) compared to the ASD-HF (M = 6.51, SE = 0.19) subgroup (F(1116) = 11.561, *p* = 0.001). The clinical data are summarized in Table 1.

In children with ASD, MRI was rated as abnormal in 64 out of 117 (54.7%), versus 14 out of 39 (35.9%) in TD children (χ^2^ = 4.1, *p* = 0.042). The ASD-HF subgroup showed an increased rate of abnormalities (65.8%; 27/41) compared to the ASD-DD subgroup (48.6%; 37/76; χ^2^ = 3.2, *p* = 0.075) and compared to the TD children (35.9%; 14/39; χ^2^ = 7.2, *p* = 0.007) IFs frequency was similar for children presenting with mild level of symptoms’ severity compared to those presenting with moderate (*p* = 0.203) or severe level (*p* = 0.128); there were no differences in IFs frequency between children presenting with moderate or with severe level (*p* = 1) (see Appendix A).

Sex differences did not account for IFs prevalence neither in ASD nor in control children. In both groups, IFs frequency was equivalently distributed between sexes: in the ASD group (HF and DD), IFs occurred in 55.5% of the males and 51.2% of the females (*p* = 0.827), while in TD children IFs were found in 33.3% of the males and 38.8%, of the females (*p* = 0.749).

Almost all TD children with a positive MRI report presented a single IF (92.8%), while this occurred for 57.8% of the children with ASD (*p* = 0.014). IFs appeared in combination in 42.2% of ASD children with positive MRI, thus in 44.4% of ASD-HF and 40.5% of ASD-DD children. A proportion of 34.4% of the ASD patients presented two IFs, while 7.8% presented 3 IFs. Table 2 shows the frequencies and type of MRI abnormalities in ASD (DD and HF) and TD groups.

Major abnormal findings among the subgroups with ASD concerned abnormal signal intensities (Figure 1a,b), among which gliosis occurred more frequently in ASD-HF compared to ASD-DD (*p* = 0.007) and ventricular anomalies prevailed in the ASD-HF subgroup compared to ASD-DD (*p* = 0.049). Ventricular enlargement and asymmetries were most frequent in ASD-HF and never observed in TD children (*p* = 0.026) (Figure 1c).

Among minor abnormal findings, mega cisterna magna appeared to be the most frequent one and was observed only in children with autism (*p* = 0.038) (Figure 1d). Virchow–Robin space enlargement was mainly found in children with ASD and was present only in 1 TD child (*p* = 0.075).

EEG reports were available for 85.5% (100/117) of the children with ASD. A total of 72% of the reports revealed a normal brain activity, while 28% were rated as abnormal. Half of the reports (14/28) documented paroxysmal anomalies (*n* = 2 were associated with epilepsy, *n* = 1 with febrile convulsions) and the other half reported as specific anomalies (*n* = 1 in association with epilepsy) (See Appendix A). Concerning the frequency of EEG anomalies or epilepsy and its association with cognitive level and/or brain IFs, EEG anomalies and epilepsy showed a similar frequency rate in both HF and DD ASD subgroups and none of them were associated with a higher presence of brain Ifs (all Ps > 0.05; See Appendix A).

Among our ASD sample, submicroscopic chromosome pathogenic abnormalities were detected in 2 children out of 117. A 15q11.2 deletion of unknown parental origin was reported in a girl with severe autism, poor language development and no intellectual deficit. EEG showed the presence of unspecific anomalies and the MRI scan revealed a choroidal cyst. 

A paternally inherited 15q13.2q13.3 microdeletion encompassing the gene CHRNA7 was detected in a boy with severe autism, absence of language and developmental delay. EEG revealed no abnormalities, while MRI showed an Arnold Chiari malformation.

## 4. Discussion

With the aim of investigating the potential utility of routine brain MRI in discovering early morphologic biomarkers for ASD, we looked at the frequency and type of IFs in a well characterized sample of non-syndromic young children with ASD compared with an age-matched sample of TD children. To our knowledge, this is the first report with detailed clinical criteria on a large sample of children with ASD aged less than 5 years. 

Previous studies investigating this issue reported contrasting data on the prevalence and nature of IFs in individuals with ASD that might have been influenced by several factors such as sex, IQ, and/or age heterogeneity, sample size, differences in scan sequences, and inconsistency in the definition of “incidental finding” as well as classification [20,24,37,38]. Differently from the reported studies, our study was stringent in the clinical characterization of the ASD sample performed by expert child neuropsychiatrists in the selection of a homogeneous and narrow age range of ASD and control groups, in the use of the same multisequence MR protocol, and in the review and interpretation of images by pediatric neuroradiologists.

Consistent with previous findings, we observed a greater occurrence of IFs in children with ASD compared to TD controls, with a frequency rate comparable to what is reported by previous studies [17,19,20,21,23]. Looking for a possible association between cognitive level and neuroradiological findings rate frequency, surprisingly, a higher rate of MRI anomalies was observed in the ASD-HF subgroup compared to both the ASD-DD subgroup and TD children. This finding is in contrast with previous studies reporting that IFs are more common in children with DD [25] or ID and in low-functioning (LF) ASD children [23]. Our results also contrast with those of Vasa et al. [24] reporting no IFs frequency differences among groups of older HF-ASD children, ADHD children and controls. However, unlike our study, none of the reported studies directly compared the IFs prevalence between two subsamples of HF and DD children with ASD. 

From a clinical perspective, our ASD-HF subsample differed from the ASD-DD both in terms of GD/IQ and of symptoms’ severity levels (see Table 1 for the distribution of our ASD individuals into mild, moderate and severe autistic symptoms categories). ASD-HF presented more frequently with “mild autistic symptoms” than the ASD-DD, although milder symptoms were not associated with a higher incidence of IFs (see Appendix A). This finding rules out the hypothesis that ASD children presenting with mild ASD symptoms constitute an outlier group driving higher IFs incidence in the ASD-HF subgroup. Consistent with the existing literature (see Appendix A), Performance IQ above 70 was chosen as the cognitive criterion for ASD-HF children. This cut-off nevertheless includes ASD children with borderline cognitive abilities who might not be the most representative individuals of an ASD-HF category. In light of this consideration, one possibility is that the higher incidence of IFs in our ASD-HF group could be ascribed to a high percentage of children with borderline IQ. Without discarding this hypothesis, a regular distribution frequency across the IQ ranges was observed among the ASD-HF children (see Appendix A). Thus, for what concerns these data, IFs incidence seems not to be correlated to the presence of individuals with borderline IQ.

In our opinion, however, those striking results do not suggest a clear pathogenic role of the IFs we found. On the other hand, we cannot exclude that the higher rate of MRI anomalies in the ASD-HF subgroup may represent a chance event due to the small sample size. Thus, further clinical MRI investigations on a larger sample size of the HF subgroup of ASD patients would be important to clarify those findings.

Notwithstanding a higher prevalence of abnormal MRI reports in ASD-HF, Fisher’s exact tests showed no difference in the frequency of observed types of IF between ASD-HF and ASD-DD children, with the exception of abnormal signal intensities (gliosis) and ventricular abnormalities, that occurred more frequently in ASD-HF children. The latter result is in accordance with the findings of Palmen et al. [39] who documented the existence of disproportional enlargements of ventricular volumes in HF patients with autism. Ventricular malformations are not always interpreted as abnormal findings [21] even if they are reported as among the most common IFs in ASD [17,40]. In the general population, the incidence of ventricular enlargement is about 1:1000 and is associated, in its most severe forms, with abnormal development, hydrocephalus and brain atrophy [41]. The presence of an enlarged ventricular volume in the ASD population has been reported in literature [42,43,44] although this feature appears to be common also in other neuropsychiatric or neurodevelopmental disorders [45,46].

In addition to ventricular abnormalities, the other most common IF among our population with ASD is represented by white matter signal abnormalities, especially in the retrotrigonal area. Blackmon and colleagues [47] reported the presence of a conspicuous volume of periventricular white matter hyperintensity in individuals with ASD as a constant anomaly over age yet associated with a higher degree of repetitive behaviors and restricted interests. Classically, this abnormality can be found in periventricular leukomalacia, metabolic disorders, viral infections or vascular disorders [48]. It has also been described in children with hydrocephalus [49]. Periventricular cerebral white matter hyperintensity has been described as the most common sign of brain injury in preterm infants, probably leading to chronic neurological morbidity [50]. The presence of areas of focal hyperintensity in deep and periventricular white matter, detectable with the T1 weighted and Flair sequences, can be indicative of gliotic areas resulting from perinatal hypoxic-ischemic insults [51]. Indeed, early white matter injury might provoke reactive astrogliosis and show as periventricular foci of T1 hyperintensity [52]. In our sample, among children with ASD presenting retrotrigonal white matter hyperintensities, 38.9% of them (7/18, 2 HF) had a history of perinatal stress. Among them, only one child presented also focal hyperintensities in the corona radiata and semioval center areas, indicating the presence of perinatal hypoxic ischemic insult.

In our study, white matter abnormalities were also represented by areas of gliosis and delayed myelination. Focal areas of gliosis in frontal, parietal and occipital white matter were observed with a significantly higher frequency in the ASD-HF sample (although only one of them presented a history of perinatal suffering), while delayed myelination was reported only in children with ASD. The association of obstetric factors with ASD was explored by Glasson and colleagues [53], reporting an increased prevalence of obstetric complications (e.g., threatened abortion, fetal distress, emergency cesarean section) among ASD children. The authors, however, excluded these factors as a single cause for autism, proposing that the increased occurrence of obstetric complications might be related to the underlying genetic factors and their interaction with the environment.

The recurrent presence of abnormal signal intensities is in accordance with the findings reported in a population of children with ASD [23] and without ID [20,21]. White matter abnormalities are not specific to ASD as they have been also described in other developmental disorders [20] and associated with motor deficits in patients with cerebral palsy [54]. However, as proposed by Boddaert and colleagues [21], the presence of white matter hyperintensities might represent the visible part of an underlying process affecting brain connectivity, in particular among areas regulating social and communication cognition, typically impaired in persons with autistic symptoms [55,56]. Aberrations in white matter tract development have been revealed by studies using the diffusion tensor imaging (DTI) techniques [57,58], further showing that the presence of an impaired white matter network in children with ASD was already evident at 6 months old [59]. In a systematic review on diffusion imaging studies in ASD children under 3 years, Conti and colleagues [60] suggested that the atypical diffusion properties of the white matter are a constant finding in the early stages of the disorder and can precede the full expression of the clinical picture.

While the investigation of white matter integrity is beyond the scope of this report, a future DTI study investigating the potential association between the presence of white matter hyperintensities and anomalies in the underlying connectivity in HF and DD-ASD during the first years of life could yield important information.

Mega cisterna magna was found in our sample with ASD only. This finding replicates those of Erbetta and colleagues [23], who documented the presence of mega cisterna magna only among children with ID and children with LF ASD. The authors proposed to consider this minor abnormality as a marker of brain dysgenesis [61]. Zimmer and colleagues [62] showed that, firstly, the enlargement of the cisterna magna is usually associated with cerebellar hypoplasia and ventriculomegaly, and secondly, that subjects presenting with isolated mega cisterna magna had a lower performance on speech task (verbal and semantic fluency), an ability often impaired in individuals with ASD. This issue could be further investigated in a subsequent study clarifying the association between the presence of mega cisterna magna and language impairments. Abnormal dilation of the cisterna magna is thought to be related to alterations in the cerebellar volumes. Developmental anomalies in the cerebellum and, in particular, hypoplasia and reduction of vermis volume have been widely reported in the past in individuals with ASD [63]. Several studies have implicated the cerebellum in the pathophysiology of autism. Recently, volumetric MRI studies have confirmed the reduction in the volume of the cerebellar worm in both HF [64] and non-HF individuals with autism [65].

Finally, the recurrence of dilated Virchow–Robin spaces in our ASD population is consistent with the literature [19,20,21,24]. In addition, an association between dilated Virchow–Robin spaces and the presence of developmental delay, psychiatric problems, and headaches was demonstrated in a pediatric population [66]. It has been proposed that their increased incidence in ID, ASD and language impairment might have clinical significance as indicating an underlying pathophysiological state [20,23]. 

The origin and significance of IFs prevalence in the pediatric population affected with neurodevelopmental disorders remain unknown and a correlation between IFs and impairments in the underlying functional and connectional architecture of the brain can only be speculated. 

The role that MRI should play in the diagnostic process of ASD is still debated. In particular, the real usefulness of classical morphological MRI is under discussion since, in most cases, it does not reveal substantial alterations in the brain structures of subjects with ASD as compared to controls. Furthermore, MRI usually reveals IFs that do not have a real clinical value, prompting clinicians to proceed with further instrumental investigations of questionable utility and high cost.

Currently, MRI investigations in individuals with ASD are not mandatory, even if they are frequently performed, and professionals in Italy often include this type of study in the ASD diagnostic process [67].

Yet, some experts believe that neuroimaging should be purely research-oriented and do not support its use in routine diagnostics. As strongly stated by Filipek et al. in the Report of the Subcommittee on Quality Standards of the American Academy of Neurology and Child Neurology Society [68], “there is no clinical evidence to support the role of routine neuroimaging in the diagnostic evaluation of autism, even in the presence of megalencephaly”. Despite these recommendations and the inherent risk of evaluating children under anesthesia, indications to carry out MRI investigations in the ASD population persist in many countries.

It must be noted that an MRI investigation could instead be justified in case of children with ASD associated with an abnormal neurological examination, even with soft neurological signs, or previous pathological findings, headaches, convulsions and with EEG abnormalities. Indeed, they show a higher prevalence of pathological findings on MRI. However, if the child has isolated ASD, MRI is unlikely to reveal a definite disease [69].

Up to date, structural and functional imaging studies have been converging on the hypothesis that ASD might be associated with atypical connectivity detectable even before the full phenotypic expression of the disorder [16,60]. Structural connectivity studies (DTI) are thus of great interest, especially when combined with studies of spatial and temporal dynamics of brain activity in specific circuits derived from fMRI, to further clarify the nature of connectivity anomalies in the brain. The combination of some of these methods within the same cohort of subjects and the application of a single study model in the early stages of ASD should be a major goal for future research [70].

## 5. Conclusions

Our results suggest that, in addition to EEG and genetic routine screenings, brain MRI may play a part in the diagnostic procedure of ASD. Yet, the data reported in our study do not allow examining to which extent the IFs prevalence in young ASD children is related to clinical and/or prognostic issues, thus the need for a routine brain MRI remains to be established. Notwithstanding, morphologic scans might be useful for unveiling the presence of IFs potentially requiring a routine follow-up, which is four-fold in ASD compared to TD children [17]. Differently from precedent studies, our discovery of IFs prevalence in ASD-HF compared to ASD-DD suggests that the presence of a developmental or intellectual delay might not constitute a strong enough criterion to perform a morphological MRI in autism, at least in very young children with ASD. This hypothesis would nonetheless need an in-depth investigation. One limitation of this study was the lack of neurobehavioral data (i.e., language development, degree of repetitive behavior and restricted interests) to correlate with the imaging findings, as it would have constituted a solid way to make a clinically meaningful interpretation of the presented data. Moreover, a possible correlation between clinical signs of autism and brain anomalies might be further unveiled by diffusion-based imaging methods and early diagnosis could benefit from prospective studies documenting brain changes over time within different developmental trajectories in ASD individuals.

## Figures and Tables

**Figure 1 brainsci-10-00741-f001:**
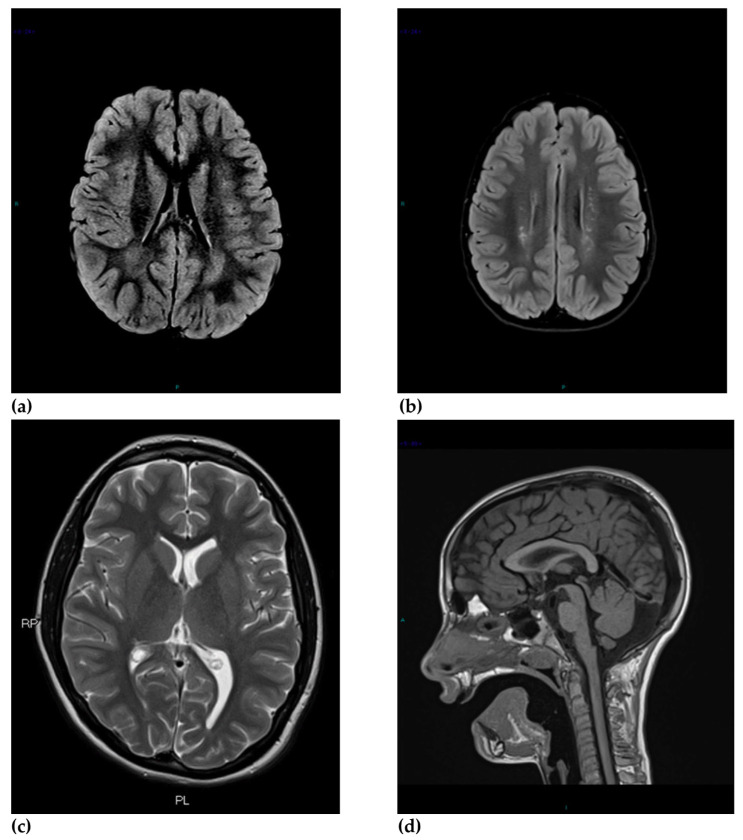
Types of MRI abnormalities in children with ASD: (**a**) White matter abnormality in children with ASD: axial FLAIR sequence in a child with ASD shows retrotrigonal white matter hyperintensity bilaterally at the posterior horns of the lateral ventricles; (**b**) White matter abnormality in children with ASD: axial T2-weight sequence in a child with ASD illustrating multiple punctuate hyperintensities scattered bilaterally in the deep white matter of the semioval centers; (**c**) Ventricular abnormality in a child with ASD: axial T2-weighted sequence shows ventricular asymmetry due to the enlargement of the posterior horn of left ventricle.; (**d**) Mega cisterna magna in a child with ASD: sagittal T1-weighted sequence shows the presence of mega cisterna magna in the posterior cranial fossa.

**Table 1 brainsci-10-00741-t001:** Clinical and sociodemographic characteristics of Children with autism spectrum disorder (ASD) and Typically Developing children.

	Children with ASD (*n* = 117)	High-Functioning ASD (*n* = 41)	Developmental Delay-ASD (*n* = 76)	Typically Developing Children (*n* = 39)	Statistics	*p*
Age (y), mean (SD), range	3.4 (0.87), 1.1–4.11	3.2 (0.85), 1.1–4.10	3.5 (0.86), 1.2–4.11	3.2(1.48), 1.1–5.7	F (2, 151) = 1509	0.224
Sex	90 M; 27 F	30 M; 11 F	60 M; 16 F	21 M; 18 F	Fisher exact test	0.008 *
ADOS-2 comparison score, mean (SD), range	7.16 (2), 3–10	6.51 (2), 3–10	7.64 (2), 4–10	NA	F (1, 116) = 11,561	0.001 *
*Mild ASD*	13	9	4	NA	Fisher exact test	0.01 *
*Moderate ASD*	46	17	29	NA	Fisher exact test	0.843
*Severe ASD*	58	15	43	NA	Fisher exact test	0.402
IQ/GD, mean (SD), range	67 (27) 25–129	94 (19) 71–129	48 (12) 25–70	NA		

Abbreviations: y, years; SD, standard deviation; F, female; M, male; IQ, intelligence quotient; GD, global development; NA, not available; * Significance level <0.05. Mild, moderate, severe ASD refers to the classification of ADOS-2’s comparative scores in three levels of symptoms’ severity.

**Table 2 brainsci-10-00741-t002:** Prevalence of MRI abnormalities in ASD and typically developing children.

MRI Findings	ASD Children (*n =* 117)	HF-ASD (*n =* 41)	DD-ASD (*n =* 76)	Typically Developing Children (*n =* 39)	*p*
Abnormal MRI	64 *	27 *	37	14 *	<0.05 *^,1^
Isolated IF	37 *	15 *	22 *	13 *	All Ps < 0.05 *
2 IFs	22	10	12	1	NS
3 IFs	5	2	3	0	NS
**Major abnormal findings**					
Abnormal signal intensities	26	10	16	4	NS
*Retrotrigonal WM intensities*	18	4	14	3	NS
*Gliosis*	7	6 *	1 *	2	0.007 *
*Myelination delay*	3	2	1	0	NS
Abnormal cortical development and/or organization	2	1	1	1	NS
Atrophy	1	1	0	0	NS
Corpus callosum anomalies	5	1	4	0	NS
Ventricular anomalies	11	7 *	4 *	3	0.049 *
*Ventricular enlargement and asymmetries*	9	6 *	3	0 *	0.026 *
*Other ventricular anomalies*	2	1	1	3	NS
Chiari I malformation	3	2	1	1	NS
Cerebellar anomalies	4	2	2	0	NS
**Minor abnormal findings**					
Dilated Virchow-Robin spaces	15	5	10	1	NS
Mega cisterna magna	12 *	5	7	0 *	0.038 *
Pineal cyst	13	4	9	2	NS
Choroid plexus cyst	2	2	0	1	NS
Arachnoid cyst	2	1	1	2	NS

Table 2 legend: “Abnormal MRI” reports the number of children with a positive MRI report. Each positive report might show a single, two or three incidental findings (IFs). The observed Ifs are categorized as major or minor abnormal findings and the table reports their frequency. Some major abnormal findings include two or more subcategories that can co-occur within the same individual, thus the category’s frequency does not necessarily correspond to the sum of its subcategories. Abbreviations: ASD, autism spectrum disorder; HF, high-functioning; DD, developmental delay; IF, Incidental Finding; Statistics: all statistics but ^1^ (Chi square test), were two-tailed Fisher exact test; * significance level <0.05.

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
