# Peer review of "Brain Magnetic Resonance Findings in 117 Children with Autism Spectrum Disorder under 5 Years Old"

_brainsci, 2020, doi:10.3390/brainsci10100741_

Round 1
Reviewer 1 Report
The authors have investigated the prevalence of incidental findings on brain MRI of a clinically derived sample of children with ASD and compared them to controls. Higher rates of IFs were found in the high IQ group with ASD as compared to controls, with the low IQ ASD group sitting somewhere in between. The methods are generally sound and the findings are of interests and add to the literature. However, the findings are of relatively modest importance being mainly a prevalence study as opposed to looking at the clinical utility of brain MRI (e.g. in identifying pathological lesions which require follow-up or as a predictor of later function).
Some specific comments and suggestions are below:
The work-up given to the children is very comprehensive and suggests that the authors practice is based in a specialist centre. In general, routine sedation and brain MRI is not employed in diagnostic work-up for uncomplicated ASD (although I do not work in the same country or healthcare system as the authors). Is the authors clinic a 'standard' one for assessment of ASD in their region or do they receive particularly complex cases?
Related to the above, how many children were diagnosed with ASD but not referred for an MRI? On what criteria were children selected for the neurological workup described?
The authors report only the rater agreement for the overall 'typical' or 'atypical' brain rating by the neuroradiologists. It would be useful to know how reliable the more specific ratings of each IF are.
The statistical analysis section states that Fisher's exact test was used to examine age differences between ASD and controls, whereas above they state they used a one-way ANOVA. This needs clarification - perhaps the Fisher's test used for gender, not age?
In supplmentary table 1 it would be useful to report the p values.
The final conclusion does not appear to me to be supported by the study - the authors state that brain MRI could play its part in the diagnostic procedure of ASD by adding data on the neurobiological substrate. However, the study they conducted did not actually examine the utility of such scans - i.e. did they reveal anything that required action / where they of prognostic significance etc. - they simply showed higher rates of incidental findings. I don't think the authors can really comment much based upon their data about the need for brain MRIs in this group, other than that they are more likely to show IFs.
Author Response
First of all, we wish to thank the reviewer for her/his in-depth reading of our paper and for the constructive attitude she/he adopted when criticizing it. We believe that her/his arguments have greatly helped us in revising the paper so as to make it clearer and, hopefully, also to make our case stronger.
Reviewer 1
1. Reviewer 1 remarked that the work-up given to the children is very comprehensive suggesting that the authors practice is based in a specialist centre. As in general, routine sedation and brain MRI is not employed in diagnostic work-up for uncomplicated ASD, she/he wondered a) whether the authors clinic is a 'standard' one for assessment of ASD in their region or do they receive particularly complex cases. Reviewer 1 further asked b) how many children were diagnosed with ASD but not referred for an MRI and questioned about c) the criteria according to which children were selected for the neurological workup described.
Answer to points a, b and c:
We agree with Reviewer 1’s comment about the MRI utility especially for “complex” ASD cases. Our Clinical Unit, one of the main regional/national reference for the diagnosis, treatment, and research on ASD, performs MRI in ASD individuals (according to parental consent) only when there are associated genetic comorbidities, EEG abnormalities/epileptic seizures, and/or neurological signs suggesting a possible cerebral lesion, alternatively when a research project requires a brain morphological study. Precisely, this very comprehensive diagnostic work-up was performed to accomplish a research project on the Genetics of Autism Spectrum Disorder aiming at discovering correlations between genotype and clinical-neurological phenotype including EEG and MRI findings. This research project is still in progress.
We modified the Material and Method part of our manuscript. Now the text contains:
P.2, l.78-88:
“188 children received a diagnosis of ASD and underwent a brain MRI in order to complete a neurological workup (mean age 5.4 y, range 13 months -28.4 y; 153 males(M)) specifically accomplishing a research project on the Genetics of ASD aimed at discovering correlations between genotype and clinical-neurological phenotype including sleep-awake EEG and MRI findings as well as standard metabolic testing, array-comparative genomic hybridization (array-CGH), and DNA analysis of MECP2 or Fragile-X syndrome (see the end of this paragraph). Exclusion criteria were as follows: defined genetic syndromes commonly associated with ASD (single gene disorders and microscopically visible chromosomal abnormalities [30]), prenatal infections including rubella and cytomegalovirus, neurometabolic or neurodegenerative diseases, severe prematurity or severe nutritional deficits and/or presence of sensory deficits, lack of parental consent to perform the MRI.”
2. Reviewer 1 asked to report the raters’ agreement for the more specific ratings of each IF.
This information has been included in the manuscript now reporting:
P.4, l.173-176:
“More specifically, a 100% agreement was present for most IFs except for the following categories: retrotrigonal WM intensities (14/18, 77.78% consensus), gliosis, (9/10, 90%), abnormal cortical development and/or organization (1/2, 50%), mega cisterna magna (11/12, 91.67%), Pineal cyst (12/13, 92.31%).”
3. Reviewer 1 asked for clarification on the statistics used to examine age differences between ASD and controls, as there was contradictory information between the statistical methods related in the text and those reported in Table 1.
This error has been corrected in the text now reporting:
P.4, l.184:
“A two-tailed Fisher’s exact test was used to examine sex differences between ASD children and controls as well as differences in the distribution of ASD-DD and ASD-HF individuals in three levels of symptoms.”
4. Reviewer 1 asked to report the p values in Supplementary table 1
Now Supplementary Table 1 reports all p values.
5. Reviewer 1 considered that the final conclusion stating that “brain MRI could play its part in the diagnostic procedure of ASD by adding data on the neurobiological substrate” could not be supported by the presented results showing higher rates of incidental findings in ASD children without examining the clinical/prognostic utility of such scans. Reviewer 1 therefore recommends interpreting the data as a report of a higher IFs prevalence in the studied group rather than data suggesting the need for brain MRIs in the diagnostic procedure of ASD individuals.
We agree with Reviewer 1’s objection, now our conclusions report the following:
P.10, l. 395-400:
“Our results suggest that, in addition to EEG and genetic routine screenings, brain MRI may play a part in the diagnostic procedure of ASD. Yet, the data reported in our study do not allow examining to which extent the IFs prevalence in young ASD children is related to clinical and/or prognostic issues, thus the need for a routine brain MRI remains to be established. Notwithstanding, morphologic scans might be useful for unveiling the presence of IFs potentially requiring a routine follow-up, which is four-fold in ASD compared to TD children [17].”
Reviewer 2 Report
This study examined the potential benefits of neuroimaging measurements across the first 5 years of life in detecting early comorbid or etiological signs of ASD. In particular the authors analyzed the prevalence of neuroradiologic findings in routine MRI scans of a group of 117 ASD children younger than 5 years old. These data were compared to those reported in TD children. MRI findings in children with ASD were analyzed in relation to their cognitive level, severity of autistic symptoms, and the presence of EEG abnormalities.
The study aimed to evaluate the effective potential of conventional MRI in unveiling early morphologic biomarkers for ASD, including IFs, in a large group of children with ASD, controlling for sample homogeneity in terms of age, developmental level, and diagnosis and compared with an age-matched control sample.
The authors state that this is the first report with detailed clinical criteria on a large sample of children with ASD aged less than 5 years. The study found that MRI was rated abnormal in 55% of children with ASD with a significant prevalence in the high-functioning subgroup compared to TD children. The study reported significant incidental findings of mega cisterna magna, ventricular anomalies and abnormal white matter signal intensity in ASD without significant associations between these MRI findings and EEG features. Based on these results the study discussed the role that brain MRI may play in the diagnostic procedure of ASD.
The study's topic is of high relevance for the field of NDD got many s. It has got many strengths. Yet further clarifications /minor revisions are needed.
Autistic symptoms severity (ADOS-2 comparative scores) was significantly higher in the ASD- DD (M=7.64, SE=0.26) compared to the ASD-HF (M=6.51, SE=0.19) subgroup (F(1,116)=11.561, P = .001).
But 104/117 had Moderate-severe ASD (classic Kanner type of autism, stable over time) that were not different in ASD-DD vs ASD-HF. Why would not consider all 104 as one group? This finding should be discussed.
Namely, only mild ASD was significantly different in ASD-DD vs ASD-HF; noteworthy that mild ASD (PDD-NOS) in 1/3 of them over time may mean no ASD, not stable.
So, authors need to elaborate on a validity of their ASD-DD vs ASD-HF definition that they used to divide categorically and interpret their intriguing MRI findings.
For example, why cut off of 70 for the above 2 groups? That is 2 SD below the mean of 100, and falls on the border of mild ID and borderline IQ.
How would then it be a helpful clinical cut off to capture ‘ ASD-HF.” Why not IQ of 80 or 85, or 90, as HF in ASD typically has higher at least verbal IQ? Thus should be included in introduction and discussed as well.
What was a frequency distribution of ASD-H for IQ 70-80, 80-90 etc?
Moreover, how did this issue affect their IQ findings, which they rely on to make this cut off? “Because of a lack of compliance of some of young patients, neurocognitive testing was done on average 16 months after the MRI assessment (between 31 months before and 106 months after the MRI examination).”
Children with ASD were diagnosed according to the DSM-5 and confirmed by both ADOS-2nd Edition and CARS-2nd Edition. For children younger than 31 months at the time of scanning, a final diagnosis was confirmed when they reached at least 31 months.
What percentage of these children below and above 31 months, and what category they fell in as for ASD-DD vs ASD-HF? In other words, how certain and reliable was the diagnosis.
Scans of all ASD and TD children were acquired on two different clinical MRI scanners: a 1.5 Tesla (n=132) or a 3T (n=24). What is a white matter resolution difference between the 1.5 T and the 3T? The majority here has had the 1.5T applied. This should be also discussed as for the study reported important periventricular white matter hyperintensity in individuals with ASD, which was followed by an informative discussion of a role of DTI etc.
More importantly, the study lacks data to correlate reported imagining findings with neurobehavioral correlates. That has been a hallmark, the only way to make a meaningful interpretation of any imaging finding.
Like, they rightfully comment that “Other studies reported of periventricular white matter hyperintensity in individuals with ASD as a constant anomaly over age yet associated with a higher degree of repetitive behaviors and restricted interests.”
The lack of correlates of the intriguing imagining findings with neurobehavioral correlates should be mentioned as a limitation.
Author Response
First of all, we wish to thank the reviewer for her/his in-depth reading of our paper and for the constructive attitude she/he adopted when criticizing it. We believe that her/his arguments have greatly helped us in revising the paper so as to make it clearer and, hopefully, also to make our case stronger.
Reviewer 2
1. In light of our results showing a higher IFs prevalence among ASD-HF children, Reviewer 2 asked for clarification on the decision to subdivide our ASD sample in two ASD-DD and ASD-HF categories according to the children’s GD/IQ level and suggested to discuss its validity in light of alternative clinical criteria such as the severity level of autistic symptomatology (ADOS-2 comparative scores).
Since almost 90% (104/117) of the whole ASD sample (i.e: ASD-DD + ASD-HF) presented moderate to severe autistic symptoms level, Reviewer 2 proposed to consider them as a pertaining to the same “Kanner type” ASD group that generally presents a stable symptomatology over time. The 13 remaining children might then constitute a separated group presenting mild autistic symptoms severity (ASD-HF) which ASD diagnosis is more likely to resolve over time and that was once categorized as PDD-NOS (see DSM-IV-TR).
We examined this possibility by subdividing our ASD group into three mild/moderate/severe ASD categories and statistically verified (Two-tailed Fisher test) whether one of those categories was differently associated with IFs. Results showed that none of these categories reported a higher IFs frequency compared to the others. Results are reported on an additional table (Supplementary Table 2) and the manuscript now reports the following:
P.4, l.186-189:
“Fisher’s exact test was also used to study the frequency of MRI abnormalities in the DD and HF subgroups compared to the TD children group and to assess the distribution of IFs in the three levels of ASD symptoms categories and in children presenting EEG anomalies”
P.5, l. 207-210:
“IFs frequency was similar for children presenting with mild level of symptoms’ severity compared to those presenting with moderate (P = .203) or severe level (P= .128); there were no differences in IFs frequency between children presenting with moderate or with severe level (P= 1) (see Supplementary Table 2).”
P. 8, l. 269-275:
“From a clinical perspective, our ASD-HF subsample differed from the ASD-DD both in terms of GD/IQ and of symptoms’ severity levels (see Table 1 for the distribution of our ASD individuals into mild, moderate and severe autistic symptoms categories). ASD-HF presented more frequently with “mild autistic symptoms” than the ASD-DD, although milder symptoms were not associated with a higher incidence of IFs (see Supplementary Table 2). This finding rules out the hypothesis that ASD children presenting with mild ASD symptoms constitute an outlier group driving higher IFs incidence in the ASD-HF subgroup.”
2. Reviewer 2 asked to elaborate on a validity of the ASD-DD vs ASD-HF definition that was used to divide categorically and interpret the MRI findings. For example, she/he asked why using a cut off of 70 for the above 2 groups that falls on the border of mild ID and borderline IQ and how would it be a helpful clinical cut off to capture ‘ASD-HF” as HF in ASD typically has higher at least verbal IQ.
We choose to use this IQ/GD cut-off according both classical references (Szatmari, 2000. “The Classification of Autism, Asperger's Syndrome, and Pervasive Developmental Disorder. Can J Psychiatry) and more recent ones (Cai et al., 2018. Increased Left Inferior Temporal Gyrus Was Found in Both Low Function Autism and High Function. Autism. Front. Psychiatry). Moreover, we have a little different point of view regarding “High functioning ASD group”; in our opinion there is a different neuropsychological profile between ASD HF and Asperger Subjects. Asperger have a higher verbal IQ while ASD HF have higher performance results. There is not a simple progressive increase of different competences in terms of IQ and verbal abilities, there is also a qualitative difference resulting possibly into two different “subgroups” with different prognosis and educational needs (see, Posar et al. 2015. “Autism according to diagnostic and statistical manual of mental disorders 5th edition: the need for further improvements”. J Ped. Neurosciences). Few previous MRI studies have been carried out to explore the different patterns of brain anatomy in children with ASD earlier than 5 years old considering different IQ levels too.
As asked by Reviewer 2, we thus introduced this issue in the Material and Method part of the manuscript:
P.3, l.101-105:
“Previous studies [20-24] demonstrated the existence of an IFs prevalence among ASD individuals with intellectual disability. Therefore, in order to verify the presence of similar data in our sample of young children with ASD, two subgroups were defined on the basis of their cognitive or developmental level. The cognitive cut-off for intellectual disability was set according to both classical and more recent references (see Appendix A).”
We reported references from the literature in Appendix A, Supplemental Methods:
P. 11, l. 441-443:
“The cut-off for DD/ID was set at a non-verbal IQ/GD score =/< 70 (i.e.: 2 SD or more below population mean) which constitutes the most commonly cut-off used in both clinical practice and research (see for example, [77-81].”
and discussed it in the Discussion part of the manuscript now reporting:
P.8, l. 275-283:
“Consistent with the existing literature (see Appendix A), Performance IQs above 70 was chosen as the cognitive criterion for ASD-HF children. This cut-off nevertheless includes ASD children with borderline cognitive abilities who might not be the most representative individuals of an ASD-HF category. In light of this consideration, one possibility is that the higher incidence of IFs in our ASD-HF group could be ascribed to a high percentage of children with borderline IQ. Without discarding this hypothesis, a regular distribution frequency across the IQ ranges was observed among the ASD-HF children (see Appendix A). Thus, for what concerns these data, IFs incidence seems not to be correlated to the presence of individuals with borderline IQ.”
3. Reviewer 2 asked for the frequency distribution of ASD-HF for IQ 70-80, 80-90
We added the data in the Appendix A, Supplementary methods:
P.11, l. 450-452:
“Concerning the distribution frequency across the IQ ranges of the ASD-HF children, 48,78% (20/41) children fell in the 71-85 GD/IQ range, 24.39% (10/41) in the 86-100 GD/IQ range, 12.20% (5/41) in the 101-115 GD/IQ range and 14.63% (6/41) in the 116-130 GD/IQ range.”
4. Reviewer 2 asked to which extent the fact that in most cases neurocognitive testing did not coincide with the time of MRI examination might have affected the IQ findings (on which the authors relied to make the ASD-HF vs DD categories.
We discussed this issue in P.12, l.453-459:
“One might question whether the temporal misalignment between neurocognitive and MRI assessments could have affected the attribution of the children to one or another ASD category (namely, ASD-HF or ASD-DD). While for the children that underwent neurocognitive testing prior to MRI, it was not always possible to access successive neurocognitive scores, if present, in order to confirm our ASD-HF/DD classification, strong evidence has shown that childhood IQ appears to remain stable over time [82]. In all cases, for which a follow up of Developmental level/IQ was available, Child Neuropsychiatrists found a stability over time.”
5. Reviewer 2 asked what percentage of the ASD children was below and above 31 months of age at the time of scanning, and what category they fell in as for ASD-DD vs ASD-HF in order to discuss how certain and reliable was the diagnosis.
We added this percentage in the Material and Method section and further developed it in the Appendix A of our manuscript:
P.3, l.95-97:
“For children younger than 31 months at the time of scanning (19/117; 16.24%), a final diagnosis was confirmed when they reached at least 31 months (see Appendix A for supplementary information).”
P.11, l.445-449:
“A proportion of 16.24%(19/117) of the children were younger than 31 months at the time of scanning (a final diagnosis of ASD was confirmed when they reached at least 31 months), 8 of them were assigned to a HF category, while the remaining 11 children have been considered as ASD-DD. The remaining 83.76% (98/117) were above 31 months at the time of the MRI assessment (65 ASD-DD and 33 ASD-HF).”
6. Reviewer 2 asked for a possible white matter resolution difference between the 1.5T and the 3T MRI scanners as most of the children participating in this study were acquired on a 1.5 Tesla and the study reported important periventricular white matter hyperintensity that were proposed to be better investigated with a DTI technique.
We added this information in the Material and Method description:
P.3, l. 129-133:
“No significant differences were found between 1.5 and 3T scanning machines yielding brain morphology assessment (in particular as concerns white matter evaluation). Namely, regardless of the machine used, the technical parameters applied (thickness, TE, TR) were set up to obtain high quality morphological sequences (T1, T2 and FLAIR T2 weighted, mainly) that provided adequate anatomical details and high contrast between tissues.”
7. Finally, Reviewer 2 asked to mention the lack of correlates of the imagining findings with neurobehavioral data as a limitation of the study, as it would have constituted a solid way to make a meaningful interpretation of the presented imaging data.
We agree with Reviewer 2’s suggestion and now our discussion is rectified with the mentioned limits of our study.
P.11, l. 404-410:
“One limitation of this study was the lack of neurobehavioral data (i.e.: language development, degree of repetitive behavior and restricted interests) to correlates with the imaging findings as it would have constituted a solid way to make a clinically meaningful interpretation of the presented data. Moreover, a possible correlation between clinical signs of autism and brain anomalies might be further unveiled by diffusion-based imaging methods and early diagnosis could benefit from prospective studies documenting brain changes over time within different developmental trajectories in ASD individuals.”
Round 2
Reviewer 1 Report
The authors have considered all relevant points in my review.